# Modulation of *Campylobacter jejuni* Motility, Adhesion to Polystyrene Surfaces, and Invasion of INT407 Cells by Quorum-Sensing Inhibition

**DOI:** 10.3390/microorganisms8010104

**Published:** 2020-01-11

**Authors:** Katarina Šimunović, Dina Ramić, Changyun Xu, Sonja Smole Možina

**Affiliations:** 1Department of Food Science and Technology, Biotechnical Faculty, University of Ljubljana, Jamnikarjeva 101, 1000 Ljubljana, Sloveniadina.ramic@bf.uni-lj.si (D.R.); 2College of Veterinary Medicine, Iowa State University, 1800 Christensen Dr, Ames, IA 50011, USA; cyxu@iastate.edu

**Keywords:** *Campylobacter jejuni*, LuxS, quorum-sensing inhibition, AI-2, natural extracts, motility, invasion of INT407 cells, adhesion to polystyrene

## Abstract

*Campylobacter jejuni* is a major foodborne pathogen, and the LuxS-mediated quorum-sensing (QS) system influences its motility, biofilm formation, invasion, host colonization, and virulence. QS therefore represents a target for the control of *C. jejuni*. The aim of this study was to investigate the correlation of QS inhibition with changes in *C. jejuni* motility, adhesion to polystyrene surfaces, and adhesion to and invasion of INT407 cells. This was achieved by studying (i) the *luxS*-deficient mutant and (ii) treatment of *C. jejuni* with 20 natural extracts as six essential oils, 11 ethanolic extracts, and three pure compounds. Compared to the wild-type, the Δ*luxS* mutant showed decreased motility, adhesion to polystyrene surfaces, and invasion of INT407 cells. The anti-QS effects of the treatments (*n* = 15/20) were assayed using *Vibrio harveyi* BB170 bioluminescence. Moderate positive correlation was shown between *C. jejuni* QS reduction and reduced motility (τ = 0.492, *p* = 0.024), adhesion to polystyrene surfaces (τ = 0.419, *p* = 0.008), and invasion (*r* = 0.394, *p* = 0.068). The best overall effect was achieved with a *Sedum rosea* (roseroot) extract, with 96% QS reduction, a 1.41 log (96%) decrease in adhesion to polystyrene surfaces, and an 82% decrease in invasion. We show that natural extracts can reduce motility, adhesion to polystyrene surfaces, and invasion of INT407 cells by *C. jejuni* through modulation of the LuxS (QS) system.

## 1. Introduction

*Campylobacter jejuni* is the leading cause of the most commonly reported bacterial gastroenteritis, campylobacteriosis, worldwide. It is considered an important food-safety hazard, and the major route for its spread is via poultry. Campylobacteriosis manifests as acute watery diarrhea, cramps, and fever, and it is associated with the development of the severe neurological condition known as Guillain-Barre syndrome [1,2,3]. The number of confirmed campylobacteriosis cases in European countries is more than 250,000 per year, and due to the overuse of antibiotics in veterinary and human medicine, more than 50% of *C. jejuni* isolates from poultry are now resistant to antibiotics [4]. The risk of resistant *C. jejuni* spreading through the food chain is therefore high, and thus the acquisition of resistant strains during infection would be more likely. Although campylobacteriosis rarely ends in death, when an infection is caused by a resistant strain, the likelihood of adverse effects increases by >5-fold [2,4,5]. Altogether, these numbers are alarming, and they represent a major burden for human health and national economies. The estimated total annual cost of campylobacteriosis and its consequences has been reported to be €2.4 billion, and it is likely to rise as the numbers of infections increase [2,6]. New targets for *C. jejuni* control and alternative options for its control are thus necessary.

The process of bacterial quorum sensing (QS) is being explored as a novel target for control of pathogens. QS is a form of cell-to-cell communication through small molecules, called autoinducers (AI), through which bacteria regulate their cell responses according to the cell density and are thus related to the behavior of the population as a whole [7,8]. In pathogenic bacteria, QS is often related to their pathogenicity. Examples of this include: *Salmonella enterica*, where QS controls the expression of virulence-associated genes and thus its virulence in mice [9]; *Escherichia coli*, where QS regulates motility, biofilm formation, and virulence [10,11]; and *Pseudomonas aeruginosa*, where QS has a vital role in biofilm formation and development of the disease cystic fibrosis [12].

Quorum sensing is regulated by signaling molecules that accumulate in the bacterial culture—specifically, *N*-acyl homoserine lactones in Gram-negative bacteria, oligopeptides in Gram-positive bacteria, and a furanosyl borate diester or autoinducer-2 (AI-2) in both Gram-negative and Gram-positive bacteria [8]. In *C. jejuni*, AI-2-mediated signaling has been described, and the role of the *luxS* gene in the synthesis of the AI-2 signal was confirmed by Elvers and Park [13]. LuxS is a *S*-ribosylhomociateinase that cleaves a toxic intermediate of the S-adenosylmethionine pathway into homocysteine and 4,5-dihydroxy-2,3-pentanedione, with the latter further spontaneously cyclized into the AI-2 signal molecule [13,14].

The phenotypic changes in the *C. jejuni luxS*-deficient mutant compared to the wild-type include weaker biofilm formation, decreased motility, higher sensitivity to oxidative stress, decreased invasion of Caco-2 cells, decreased virulence in a guinea pig abortion model, and decreased colonization of the chicken intestine [13,15,16,17,18,19,20]. As the *C. jejuni* QS-deficient mutant shows critical deficiencies in colonization and virulence, QS inhibition in *C. jejuni* can be considered a major target for the control of this pathogen in the host.

Bacterial QS can be inhibited by natural extracts and their major pure compounds [8]. For example, garlic extracts reduce *Psuedomonas aeruginosa* biofilm formation and virulence in vitro and in vivo in a mouse pulmonary infection model [21,22]. Specifically, ajoene is a sulphur-rich compound that was isolated from this garlic extract and is considered to be responsible for this activity. Ajoene shows high anti-QS activity and reduces *P. aeruginosa* infection in a mouse model, similar to the crude extract [23]. Similarly, rutin is a flavonoid found in various plants, such as olives, buckwheat, and raspberries, and it can reduce biofilm formation and attenuate virulence of pathogenic *E. coli* through inhibition of QS [24].

Quorum-sensing inhibition by phytochemicals and its consequences in *C. jejuni* have been poorly studied. It is believed that citrus extracts reduce motility, biofilm formation, invasion, and adhesion of epithelial cells and virulence of *C. jejuni* by modulation of QS [25,26]. A *Euodia ruticarpa* extract has also shown anti-QS activity, although a link between reduction of biofilm formation and QS activity was not shown [27].

Although QS is a prominent target for pathogen control, it is hard to speculate what the effects of its modulation by outside sources would be, such as for phytochemical inhibitors of QS in *C. jejuni*. The *C. jejuni* phenotype that is associated with inhibited QS has been described through studies of *luxS*-deficient mutants, although it is not clear which of its features are caused by lack of cell-to-cell communication and which are caused by disruption of the S-adenosylmethionine metabolic pathway.

The aim of the present study was to determine the influence of QS inhibition on *C. jejuni* motility, adhesion to polystyrene surfaces, and adhesion to and invasion of INT407 epithelial cells. For this purpose, we chose 20 treatments with essential oils, ethanolic extracts, and pure compounds. The changes in the phenotype of the wild-type *C. jejuni* 11168 under these treatments were compared to the changes that occurred in the *C. jejuni* Δ*luxS* mutant to effectively rule out phenotypic changes due to the influence of the treatments on *C. jejuni* systems other than LuxS.

## 2. Materials and Methods

### 2.1. Bacterial Strains and Growth Conditions

*Campylobacter jejuni* NCTC 11168 and *C. jejuni* 11168Δ*luxS* [19,27] provided by Klančnik A. and Bezek K. (University of Ljubljana) were stored at −80 °C in 20% glycerol and 80% Mueller–Hinton broth (MHB; Oxoid, UK). They were grown on Mueller–Hinton agar (MHA; Oxoid, UK) at 42 °C under microaerobic conditions (5% O_2_, 10% CO_2_, 85% N_2_) for 24 h. The second passage from each culture was used in the experiments. When necessary, MHA was supplemented with selective medium (SR01176; Oxoid, UK) and growth medium (SR0232E; Oxoid, UK) (MHA-SS) or 30 mg/L kanamycin (Merck, Darmstadt, Germany). The *Vibrio harveyi* BB170 reporter strain [27,28] was grown on autoinducer bioassay (AB) medium at 30 °C, which contained 17 g/L NaCl (Merck, Darmstadt, Germany), 12.3 g/L MgSO_4_ (Merck, Darmstadt, Germany), 2 g/L casamino acids (BD Bacto, Fisher Scientific, Hampton, NH, USA), 1 mM K_2_HPO_4_ (Kemika, Zagreb, Croatia), 0.1 mM L-arginine (Sigma Aldrich, Steinheim am Albuch, Germany), and 1% (*v*/*v*) glycerol (Kemika, Zagreb, Croatia).

### 2.2. Essential Oils and Pure Compounds

All of the essential oils were commercially obtained from companies in Slovenia, as: juniper (*Juniperus communis*; EO1; from Herbana d.o.o.); oregano (*Origanum sp*.; EO2); cloves (*Syzygium aromaticum*; EO3); rosemary (*Rosmarinus officinalis*; EO4); and thyme (*Thymus vulgaris*; EO6; all from Lek Veterina d.o.o.); and lavender (*Lavandula hybrida*; EO5; from M. Jeršek s.p.) (Appendix A).

The pure compounds used in this study were: carvacrol (P1); rosmarinic acid (P2); and *γ*-terpinene (P3), and they were all from Sigma Aldrich (Steinheim am Albuch, Germany) (Appendix A).

### 2.3. Extract Preparation

Various ethanolic extracts were used in this study for the treatments, and these are described in Appendix A. These were prepared from the following plants (Kottas Pharma, Vienna, Austria): oregano (*Origanum* sp.), predistillation (E1) and post-distillation (E2), and from leaves (E3) and flowers (E4) separately; nettle (*Urtica dioica*), predistillation (E5) and post-distillation (E6); winter savory (*Satureja montana*) (E7); roseroot (*Sedum rosea* prev. *Rhodiola rosea*) (E8); yarrow (*Achillea millefolium*), predistillation (E9) and post-distillation (E10); and rosemary (*Rosmarinus officinalis*) (E11).

The plant materials were dried and ground before the ethanolic extractions with 150 mL 96% denatured ethanol (Roth, Germany) added to 20 g dried material. The suspensions were heated until boiling, stirred for 30 min, and filtered (Rotilabo pleated paper filters; Roth, Germany) to remove undissolved particles. The solvent was evaporated off using a rotary evaporator (Rotavapor; Büchi, Flawil, Switzerland) at 40 °C and 175 bar. The post-distillation material was prepared with the same protocol but used after the distillation process. Finally, extracts were dried fully using a nitrogen flow and stored at −20 °C. The roseroot (E8) was provided by Bucar F. (Department of Pharmacognosy, Institute of Pharmaceutical Sciences, University of Graz, Graz, Austria) and was prepared as described by Alperth et al. [29].

### 2.4. Determination of Minimal Inhibitory Concentrations

The minimal inhibitory concentrations (MICs) against *C. jejuni* of all of the treatments were initially determined using the broth microdilution method, as described previously [30]. For this purpose, all stock solutions were prepared in dimethylsulphoxide (DMSO) at the stock concentration of 40 mg/mL and then further diluted in MHB. The final concentration of DMSO used in the assays did not exceed 2.5% in MHB with appropriate DMSO controls also included. Treatments with 0.25× MICs were chosen for the AI-2 QS inhibition bioassays (Appendix A).

### 2.5. Autoinducer-2 Bioassay

To determine the influence of these treatments on *C. jejuni* QS, AI-2 bioassays were performed. The changes in *C. jejuni* AI-2 activities or QS were inferred from the changes in the measured luminescence of the *V. harveyi* BB170 reporter strain in relative luminescent units (RLU).

*Campylobacter jejuni* 11168 and *C. jejuni* 11168Δ*luxS* [19] cultures in MHB were adjusted to OD_600_ 0.1 (5 × 10^7^ CFU/mL). Here, the treatment stock solutions were prepared as 100× the concentrations of 0.25× MIC for each treatment in 100% DMSO. Then, 50 µL of each treatment stock was added to 5 mL *C. jejuni* cultures (at OD_600_ 0.1) for the final concentration of 0.25× MIC. Cultures treated with 1% DMSO were used as controls. The cultures were incubated under microaerobic conditions at 42 °C for 24 h. To obtain the cell-free supernatants (CFSs) for the following bioassay, samples of 2 mL were taken and filter sterilized using 0.2 µm syringe filters (Sartorius, Göttingen, Germany).

The AI-2 bioassays were performed as previously described [27,28] with some modifications. These QS inhibition bioassays were carried out using the biosensor reporter strain *V. harveyi* BB170 [27], which was grown for 16 h at 30 °C and 150 rpm mixing to approximately 10^7^ CFU/mL and used at the final concentration of 5 × 10^3^ CFU/mL in AB medium. The *C. jejuni* CFSs were added to the suspensions of the reporter strain to a final concentration of 10% (*v*/*v*) (i.e., 20 µL CFS added to 180 µL reporter strain suspension). Sterile medium was used as the blank [10% (*v*/*v*) MHB, 90% (*v*/*v*) AB medium] and the negative control was 10% (*v*/*v*) MHB and 90% (*v*/*v*) *Vibrio harveyi* suspension. Kinetic measurements were carried out for the bioluminescence signals of the *V. harveyi* BB170 that were produced as a result of the QS signal that originated from the presence of the CFSs. The relative luminescence signals were measured at 15 min intervals over 20 h at 30 °C in white microtiter plates (Nunc, Thermo Fisher Scientific, Roskilde, Denmark incubated in a microplate reader (Varioskan Lux; Thermo Scientific, Waltham, MA, USA). The relative luminescence signals were interpreted as the QS signal in the *C. jejuni* CFSs (i.e., higher signals indicated higher concentrations of QS signaling molecules in the CFSs produced by *C. jejuni*).

When added to MHB, *V. harveyi* produces a background luminescence signal that increases with the concentration of the culture. To find the most stable point of signal production, the CFSs from *C. jejuni* 11168∆*luxS*, a mutant that cannot produce the quorum-sensing signal AI-2, and fresh MHB were used as the negative controls. The time point when the *V. harveyi* signal production after addition of MHB and *C. jejuni* 11168Δ*luxS* was still the same but differed from *C. jejuni* 11168 CFSs (i.e., after 14.5 h incubation) was used for the calculation of the QS signals attributed to *C. jejuni* (Appendix A).

The experiments were performed as three independent biological replicates and three technical replicates. The data presented are means ± standard deviation from three biological replicates.

### 2.6. Motility Assay

*Campylobacter jejuni* 11168 and *C. jejuni* 11168Δ*luxS* were resuspended in MH broth, and the OD_600_ was adjusted to 0.1. The treatments were added to these *C. jejuni* liquid cultures to the final concentrations of 0.25× MIC in 1% DMSO. Control cultures were treated with 1% DMSO. The prepared cultures were incubated at 42 °C under microaerobic conditions for 16 h. After these incubations, 1 μL of each culture was dropped onto 0.4% MH agar and incubated at 42 °C under microaerophilic conditions for 72 h. The diameter of each colony that formed was measured in mm. The experiments were carried out as three technical replicates, where one plate represented one technical replicate, and in three biological replicates on separate occasions. Data are means ± standard deviation of the biological replicates. The differences in the colony diameters (mm) were determined, and the effectiveness of the treatments on *C. jejuni* motility inferred from the changes in colony diameters (Δmm).

### 2.7. Adhesion to Polystyrene Surfaces

The treatments were carried out in liquid cultures of *C. jejuni* 11168 and *C. jejuni* 11168Δ*luxS* (at OD_600_ 0.1) at the final concentrations of 0.25× MIC and 1% DMSO in MHB. Control cultures were treated with 1% DMSO. Then, 100 μL of the prepared cultures were added to a 96-well polystyrene microtiter plate (untreated). The plate was incubated at 42 °C under microaerophilic conditions for 24 h. After the incubation, the wells were washed with phosphate-buffered saline (PBS) three times and treated in an ultrasonic bath (Iskra Pio, Šentjernej, Slovenia) for 10 min. Then, 10-fold serial dilutions were plated onto MH agar. The numbers of adhered *C. jejuni* were determined as CFU/well. The experiments were carried out in three technical replicates, where one well was one technical replicate, and in three biological replicates on separate occasions. Data are means ± standard deviation of the biological replicates. The effectiveness of the treatments on *C. jejuni* adhesion was inferred from the differences between the adhered *C. jejuni* (CFU/well) of the treated cultures compared to the control.

### 2.8. Invasion and Adhesion of INT407 Cells

The adherence to and invasion of human intestinal epithelial cells by *C. jejuni* 11168 and *C. jejuni* 11168Δ*luxS* after the treatments were determined using the INT407 cell line (ATCC), according to Negretti and Konkel [31], with some modifications. The effectiveness of the treatments for *C. jejuni* QS inhibition was inferred from the change in adhered/invaded *C. jejuni* (CFU/mL) in the INT407 cell cultures compared to the untreated control.

The INT407 cells were cultured in Dulbecco’s modified Eagle’s medium (DMEM; Corning, Mediatech Inc., Manassas, NA, USA) supplemented with 1% HEPES (Gipco, Life Technologies, Grand Island, NY, USA), 1% glutamax (Gipco, Life Technologies, Grand Island, NY, USA), and 10% fetal bovine serum (FBS) (Sigma, Darmstadt, Germany) in 75 cm^2^ culture flasks (Corning, Corning, NY, USA) at 37 °C in 5% CO_2_. The INT407 cells were trypsinized (0.25% trypsin; Corning, Corning, VA, USA) and adjusted to 1.5 × 10^5^ cells/mL. One milliliter of the cell suspensions was added to each well of a 24-well microtiter plate (Corning, Corning, NY, USA) and left overnight at 37 °C under 5% CO_2_.

For the bacterial culture preparation, *C. jejuni* 11168 and *C. jejuni* 11168Δ*luxS* were suspended in MH broth, and the OD_600_ was adjusted to 0.1. Extracts were added to liquid cultures at the final concentration of 0.25× MIC in 1% DMSO. Control cultures were treated with 1% DMSO. The cultures were then incubated at 42 °C under microaerobic conditions for 16 h. After the incubations, the cultures were centrifuged (6000× *g*, 20 °C) and washed twice with PBS. The final cultures were resuspended in DMEM1% (DMEM with 1% fetal bovine serum, 1% HEPES, 1% glutamax), and the OD_600_ adjusted to 0.3. These suspensions were then diluted 1:10 in DMEM1% to reach an OD_600_ of 0.03 or 3 × 10^7^ CFU/mL and were then used to infect the INT407 cells.

The plates with the INT407 cells were washed twice with DMEM1%, inoculated with 1 mL of the prepared *C. jejuni* suspensions, and incubated 3 h at 37 °C under 5% CO_2_. For determination of the total *C. jejuni*, the plates were washed three times with PBS, and the INT407 cells were lysed with 200 μL 0.1% Triton X-100 (Sigma, MD, USA) for 5 min at 37 °C. Following the incubation, 800 μL PBS was added to each well, and to determine the CFU/mL of *C. jejuni*, 10-fold serial dilutions were performed with plating onto MHA plates and incubation at 42 °C under microaerobic conditions.

For determination of the internalized *C. jejuni*, the plates were washed twice with DMEM1%, and 1 mL DMEM1% supplemented with 100 mg/L gentamycin (Gipco, Life Technologies, Grand Island, NY, USA) was added. The plates were incubated for an additional 3 h at 37 °C in 5% CO_2_. The plates were then washed, the INT407 cells lysed, and the numbers of internalized *C. jejuni* determined as described above. The number of adhered cells was determined by subtracting the number of internalized cells from the total *C. jejuni* concentration. The concentrations of the adhered and the internalized *C. jejuni* are reported as log_10_ CFU/mL.

The experiments were carried out in three technical replicates, where one replicate was one well in the plate, and in three biological replicates on separate occasions. Data are means ± standard deviation of the biological replicates.

### 2.9. Statistical Analysis

For all the data, before statistical tests were performed, normality was tested with Shapiro–Wilk and Kolmogorov–Smirnov tests. All the CFU/mL and the RLU values were log transformed before analysis. To determine the statistical significance between treated and control cultures, Student’s t-tests or Mann–Whitney U-tests were performed, depending on the normality distribution of the data. For correlation analysis, only mean values and not replicates were used for each treatment. Correlations between quorum sensing (luminescence), invasion and adhesion (log_10_ CFU/mL), motility (diameter), and adhesion to polystyrene surfaces (log_10_ CFU/well) were tested using Pearson’s or Kendall’s Tau correlation tests, depending on the normality distribution of the data.

All the analyses were performed using the SPSS software, version 22 (IBM Corp., Armonk, NY, USA).

## 3. Results

### 3.1. Comparison of Wild-Type C. jejuni 11168 with the luxS-Deficient Mutant

To evaluate the influence of QS inhibition caused by the loss of the *luxS* gene that is responsible for AI-2 signal production in *C. jejuni*, wild-type *C. jejuni* 11168 and the *luxS*-deficient mutant were tested for motility on soft agar, adhesion to polystyrene surfaces, and adhesion to and invasion of INT407 cells.

Compared to the wild-type *C. jejuni* 11168, the *C. jejuni* 11168 Δ*luxS* mutant strain showed a small but significant decrease in motility of 4.7% (3.67 mm; *p* = 0.018) and significantly lower adherence to polystyrene surfaces of 98.8% (1.82-log units; *p* < 0.001), and although adhesion on INT407 cells was not significantly altered, the invasion of the INT407 cells by the *C. jejuni* 11168Δ*luxS* mutant was significantly lower than for the wild-type by 1.10 log units (92.1%; *p* = 0.003) (Figure 1).

### 3.2. Inhibition of Quorum Sensing of C. jejuni with Essential Oils, Ethanolic Extracts, and Pure Compounds

Inhibition of *C. jejuni* QS was examined using the AI-2 bioluminescence assays, where the *C. jejuni* CFSs, which contained the QS signals, were added to the biosensory strain *V. harveyi* BB170. Reduction of *C. jejuni* QS by these essential oils, ethanolic extracts, and pure compounds was seen as a decrease in the luminescence signal where the CFSs of treated cultures were added in comparison to the control CFS.

Most of these treatments (*n* = 15/20) significantly lowered the *C. jejuni* QS signals compared to the control (Figure 2). The luminescence signal decreases ranged from 27% to 96% (Table 1) with the strongest effect achieved by the ethanolic extract E8 (roseroot). One of the essential oils (EO2, oregano), two of the ethanolic extracts (E1, oregano, predistilled; E11, rosemary), and two of the pure compounds (P1, carvacrol; P2, rosmarinic acid) did not have any significant impacts on *C. jejuni* QS.

### 3.3. Modulation of C. jejuni Motility

The *C. jejuni* 11168 and the *C. jejuni* 11168Δ*luxS* motilities were examined in cultures on soft agar that was treated with these essential oils, ethanolic extracts, and pure compounds. The motility of the *C. jejuni* wild-type was significantly reduced by three of the essential oils (EO1, juniper; EO3, cloves; EO5, lavender), three of the ethanolic extracts (E4, oregano, flowers; E6, nettle, post-distillation; E8, roseroot), and two of the pure compounds (P2, rosmarinic acid; P3, γ-terpinene). The E8 treatment (*S. rosea*, roseroot) had the strongest effect, as it decreased the *C. jejuni* 11168 colonies by 27 mm (35%) (Figure 3A). Although statistical significance was not reached due to the high variability between the experiments here, the motility appeared to be decreased by >15% also by essential oils EO2 (oregano) and EO4 (rosemary) and by ethanolic extract E11 (rosemary) (Figure 3A, Table 1, Appendix A).

Unlike the wild-type, the motility of the Δ*luxS* mutant strain was not significantly changed by the treatments compared to the control; overall, none of these treatments showed any reduction of the motility of *C. jejuni* 11168Δ*luxS* by >10% (Appendix A).

### 3.4. Modulation of C. jejuni Adhesion to Polystyrene Surfaces

The number of cultivable *C. jejuni* 11168 and *C. jejuni* 11168Δ*luxS* that were adhered to a polystyrene surface after a 24 h incubation with or without these treatments with essential oils, ethanolic extracts, and pure compounds (at 0.25× MIC) were determined as a measure of the *C. jejuni* biofilm formation on polystyrene surfaces. The adhesion of the wild-type was significantly reduced by nine of these treatments (*n* = 9/20): four essential oils (EO1, juniper; EO3, cloves; EO4, rosemary; EO5, lavender); four ethanolic extracts (E2, oregano, post-distilled; E3, oregano, leaves; E4, oregano, flowers; E8, roseroot); and one of the pure compounds (P3, γ-terpinene). The reductions caused by the treatments ranged from 0.13 to 1.89 mean log units (26% to 99%) (Figure 3B, Table 1, Appendix A), where the greatest reduction was achieved by essential oil EO1, from juniper. The treatments showed no significant effect on *C. jejuni* 11168Δ*luxS* adhesion to polystyrene surfaces (Appendix A).

### 3.5. Modulation of Adhesion to and Invasion of INT407 Cells by C. jejuni 11168

The effects of these treatments of *C. jejuni* 11168 and 11168Δ*luxS* on their adhesion to and invasion of INT407 cells was investigated. *C. jejuni* cultures were incubated together with these treatments (at 0.25× MIC) for 16 h, then washed and added to the INT407 epithelial cells. In this way, only treated *C. jejuni* were added without transfer of essential oils, ethanolic extracts, or pure compounds to the INT407 cells. Thus, the changes in adhesion or invasion can be considered to result from phenotypic changes of the *C. jejuni* cultures due to these treatments.

The adhesion of the treated *C. jejuni* 11168 cultures to the INT407 cells did not significantly differ from the control for the majority of these treatments (*n* = 17/20). Indeed, only treatments E4 (oregano, flowers), E7 (winter savory), and P3 (γ-terpinene) showed significantly lower *C. jejuni* adhesion by 0.67, 0.58, and 0.62 mean log units, respectively (Figure 3C, Appendix A). The *C. jejuni* 11168Δ*luxS* adhesion to INT407 cells was not influenced by any of these treatments (Appendix A).

The invasion of the INT407 cells by *C. jejuni* 11168 was, on the other hand, significantly modified by most of these treatments (*n* = 15/20; Figure 3D). The reduced invasion ranged from 0.57 to 1.02 mean log units, with P1 (carvacrol) showing the greatest effect (91%). The five of these 20 treatments that had no significant effects on invasion were four ethanolic extracts (E4, oregano flowers; E5, nettle pre-distilled; E6, nettle post-distilled; and E10, yarrow pre-distilled) and one pure compound (P2, rosmarinic acid) (Figure 3D, Table 1). The invasion of *C. jejuni* 11168Δ*luxS* was not affected by these treatments (Appendix A).

It should be noted that none of the treatments achieved the same effects on invasion reduction as did the knock-out mutation of the *luxS* gene.

### 3.6. Correlation Analysis

The correlations between the QS reductions of the *C. jejuni* (luminescence signal, logRFU) with the motility (diameter, mm), the adhesion to and invasion of INT407 cells (logCFU/mL), and the adhesion to polystyrene surfaces (logCFU/mL) after these treatments with essential oils, ethanolic extracts, and pure compounds were examined. Only means for luminescence, colony diameters, and *C. jejuni* logCFU/well were used here. In this way, we determined the correlation of treated cultures without the noise from the error between the experiments.

This analysis showed significant, although moderate, positive correlations of luminescence (i.e., QS) with *C. jejuni* 11168 motility (τ = 0.492, *p* = 0.024) and adhesion to polystyrene surfaces (τ = 0.419, *p* = 0.008). These suggest that the decrease in luminescence, and thus the *C. jejuni* QS, results in decreased *C. jejuni* motility on soft agar and adhesion to a polystyrene surface.

These data also showed moderate positive correlation of luminescence and invasion of the INT407 cells (*r* = 0.394), although this remained a trend as statistical significance was not reached (*p* = 0.068). However, this might suggest an influence of the reduced *C. jejuni* QS on their invasion of these INT407 epithelial cells.

## 4. Discussion

The LuxS-mediated cell-to-cell signaling system, and thus QS, has an important role in *C. jejuni* motility, biofilm formation, host colonization, and virulence [13,15,16,18,19].

In the present study, we investigated the modulation of the *C. jejuni* QS system by (i) knockout mutation of the *luxS* gene, which is responsible for AI-2 signal production, and (ii) treatments with 20 natural extracts (essential oils, ethanolic extracts, pure compounds). We studied the changes that these modulations might induce in terms of the *C. jejuni* motility on soft agar, adhesion to polystyrene surfaces, and adhesion to and invasion of human INT407 epithelial cells.

The *luxS*-deficient mutant showed statistically significant reduction in motility on soft agar (4.7%), adhesion to polystyrene surfaces (1.47-log reduction), and invasion of INT407 cells (1.10-log reduction) compared to the wild-type *C. jejuni*. Decreased motility of the *luxS*-deficient mutant compared to the wild-type was also reported by Elvers and Park [13], Jeon et al. [16], Holmes et al. [17], Quiñones et al. [18], and Plummer et al. [19]. Furthermore, Jeon et al. [16] and Holmes et al. [17] showed that the expression of the *flaA* gene, which is crucial for flagella-mediated motility, is significantly reduced in the *luxS*-deficient mutant. Elvers and Park [13] also showed somewhat weaker invasion of Caco-2 cells by the *luxS*-deficient mutant compared to the wild-type (37.5% lower) and no effect on adhesion. Quiñones et al. [18], on the other hand, showed significant reduction in adhesion to Leghorn male hepatoma (chicken hepatoma) cells by the *luxS* mutant compared to the wild-type.

The response of *C. jejuni* 11168 and the knockout mutant *C. jejuni* 11168Δ*luxS* to 20 natural formulations, namely six essential oils, 11 ethanolic extracts, and three pure compounds that are often found in plant materials, were compared using the concentration of 0.25× MIC to ensure they did not influence *C. jejuni* growth.

The effects of these treatments on the *C. jejuni* LuxS system were measured with the bioluminescence assay, whereby a reduction in the *C. jejuni* QS can be inferred from a reduction in the luminescence measured. Significant *C. jejuni* QS reduction was achieved by the majority of these treatments (*n* = 15/20; Figure 2). Indeed, a number of these treatments have been previously investigated as QS inhibitors in various organisms, although not in *C. jejuni*. Juniper [32], lavender [33], cloves [34], thyme [35], and oregano [36] have been reported to show anti-QS activities, which we confirmed here in *C. jejuni*. Nettle [37] and yarrow [38] had no effects on *E. coli* QS and *Chromobacterium violaceum*, respectively, although in the present study they reduced the AI-2 levels (and thus QS) in *C. jejuni*. In *C. violaceum*, carvacrol has also been reported to have anti-QS activity [39], although this was not shown here for *C. jejuni* QS.

Castillo et al. [25] reported that citrus extracts reduced *C. jejuni* AI-2 activity in this bioluminescence assay, which correlated with its motility and biofilm formation. Furthermore, these citrus extracts reduced *C. jejuni* invasion of and adhesion to HeLa cells and the expression of the virulence genes *cadF* and *ciaB* [26]. On the other hand, Bezek et al. [27] showed good *C. jejuni* QS reduction with *Euodia ruticarpa* but no connection to *C. jejuni* biofilm formation. This indicates high sensitivity of the *C. jejuni* LuxS system, and thus QS signaling, to outside stressors, such as treatments with natural extracts. However, it does not conclusively confirm the effects of *C. jejuni* QS inhibition on the other phenotypic changes in *C. jejuni*.

We showed here that the motility of *C. jejuni* 11168 was significantly reduced by only eight of these 20 treatments, while the motility of *C. jejuni* 11168Δ*luxS* was not affected by any of them. Interestingly, the greatest effect on motility was achieved by the *S. rosea* (roseroot) ethanolic extract (E8), which also had the largest effect on *C. jejuni* QS. Indeed, a moderate but significant positive correlation of the luminescence, and thus QS, and *C. jejuni* 11168 motility was found after these treatments (τ = 0.492, *p* = 0.024). This suggests that treatments with these essential oils, ethanolic extracts, and pure compounds can modulate *C. jejuni* motility through modulation of the LuxS/QS system. *C. jejuni* motility is a promising target for reduction of *C. jejuni* host colonization, as a loss of motility greatly reduces its colonization [40,41].

Flagella-mediated motility and QS also have important roles in *C. jejuni* biofilm formation, because non-motile mutants and *luxS*-deficient mutants have shown weaker biofilm formation compared to the wild-type [42,43]. As a measure of *C. jejuni* biofilm formation on polystyrene surfaces, we determined here the number of cultivable *C. jejuni* that adhered to the walls of a polystyrene plate. The adhesion of *C. jejuni* 11168 to the polystyrene surface was significantly reduced by nine of these treatments with essential oils, ethanolic extracts, and pure compounds. The juniper essential oil (EO1) had the strongest effect here, as it reduced *C. jejuni* adhesion by 1.89-log units (approximately 99%). The same strong anti-adhesion effect of juniper essential oil (99% reduction) was also demonstrated by Klančnik et al. [44] using qPCR quantification of *C. jejuni* adhesion.

Significant moderate positive correlation was found here between luminescence (i.e., QS) and adhesion of *C. jejuni* to polystyrene surfaces after these treatments. This thus indicates that reduced luminescence as AI-2 activity for *C. jejuni* corresponds to reduced adhesion.

The adhesion of the *luxS* mutant was lower compared to the wild-type, which again suggests involvement of the LuxS system of *C. jejuni* in its adhesion to polystyrene surfaces. Consequently, the modulation of the LuxS system results in reduced adhesion of *C. jejuni*.

The influence in the present study of these treatments with essential oils, ethanolic extracts, and pure compounds on *C. jejuni* invasion of and adhesion to INT407 epithelial cells was tested. Only three of these treatments of *C. jejuni* 11168 reduced their adhesion to the INT407 cells (E4, oregano, flowers; E7, winter savory; P3, γ-terpinene), and none reduced the adhesion of *C. jejuni* 11168Δ*luxS*. Although the adhesion of *C. jejuni* to the INT407 epithelial cells was generally not affected, the majority of these treatments (*n* = 15/20) reduced the *C. jejuni* 11168 invasion of the INT407 cells. None of the treatments achieved the same level of effect as the knock-out mutation of the *luxS* gene. This was not surprising, as none of the treatments reduced the *C. jejuni* QS signal by 100%. The effects of these treatments on *C. jejuni* 11168 and the absence of effects on *C. jejuni* 11168Δ*luxS* show that modulation of the LuxS system by outside stressors can result in changed *C. jejuni* invasiveness, which in this case was caused by most of these treatments with essential oils, ethanolic extracts, and pure compounds. Furthermore, a moderate, although not statistically significant, positive correlation was shown between *C. jejuni* AI-2 activity and *C. jejuni* invasiveness of the INT407 epithelial cells (*r* = 0.394, *p* = 0.068). Lower cell invasiveness of the *luxS* mutant on Caco-2 cells and no effect on adhesion were previously reported by Elvers and Park [13]. This shows the sensitivity of *C. jejuni* to the loss or inhibition of the QS system in terms of epithelial cell invasion reduction but not adhesion.

The treatment that showed the greatest *C. jejuni* QS reduction was the *S. rosea* (roseroot) ethanolic extract (E8), and this also had the greatest effect on *C. jejuni* motility and was among the best in terms of reduced invasion of INT407 cells and adhesion to polystyrene surfaces by *C. jejuni*. The roseroot extract is composed of different cyanogenic glycosides, phenylethanoids, procyanidin and catechin derivatives, phenylpropanoids, monoterpene alcohols, and flavonoids [29], and isolated compounds of this extract should be considered in the search for new anti-QS compounds against *C. jejuni.* Also, rosmarinic acid (P2) was among the weakest in terms of effects on *C. jejuni* QS, motility, and adhesion to polystyrene surfaces. The moderate positive correlation of the *C. jejuni* QS signal activity with *C. jejuni* invasion of epithelial cells, motility, and adhesion to polystyrene surfaces with these treatments and the absence of any effects with the Δ*luxS* mutant indicate a role for the LuxS system in the regulation of these phenotypes. This now defines the LuxS system as an interesting target for the control of *C. jejuni* and also that such plant-derived substances are potential QS inhibitors and thus potential anti-*Campylobacter* agents.

## 5. Conclusions

The beneficial effects of used treatments against the wild-type *C. jejuni* and the lack of effect against the *luxS*-deficient mutant strongly suggest the influence of these treatments on the LuxS system. The positive, although moderate, correlation of QS signal reduction with reduced invasion of epithelial cells, motility, and adhesion to polystyrene surfaces links these phenotypic changes. However, as not all the changes in QS seen here resulted in the changed phenotypes, other mechanisms of action of such plant extracts remain to be considered before their use for reduction of *C. jejuni* in vivo.

We confirmed here that modulation of the *C. jejuni* LuxS system with essential oils, ethanolic extracts, and pure compounds can result in reduced *C. jejuni* invasion of INT407 epithelial cells and motility and adhesion to polystyrene surfaces, and that these phenotypic changes correlate with the measured AI-2 activity. To confirm the anti-QS effects of these treatments, further studies are warranted in *C. jejuni*. Better quantitative measurement systems and the complementation of the *C. jejuni luxS* mutant as well as the addition of a pure AI-2 control would further confirm that these natural plant-derived essential oils, ethanolic extracts, and pure compounds act as true QS modulators.

## Figures and Tables

**Figure 1 microorganisms-08-00104-f001:**
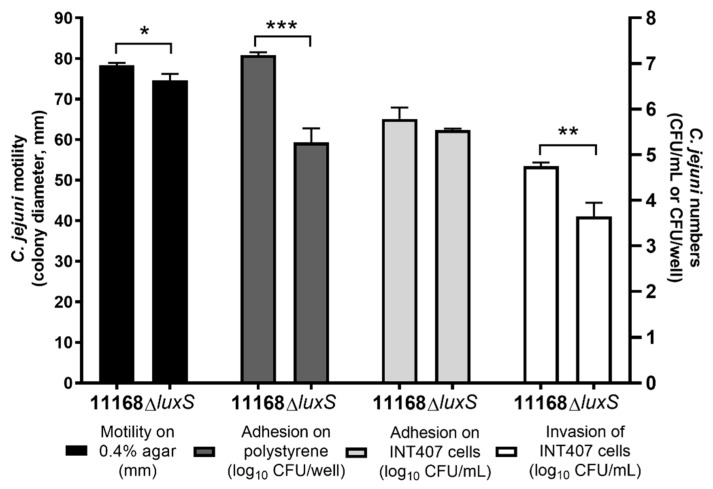
*C. jejuni* 11168 and *C. jejuni* 11168Δ*luxS* motility presented as colony diameters (mm), adhesion on polystyrene surfaces presented as log_10_ CFU/well, and adhesion to and invasion of INT407 cells presented as log_10_ CFU/mL. Data are means ± standard deviation. * *p* < 0.05; ** *p* < 0.01; *** *p* < 0.001, Δ*luxS versus* 11168 control (Student’s *t*-tests).

**Figure 2 microorganisms-08-00104-f002:**
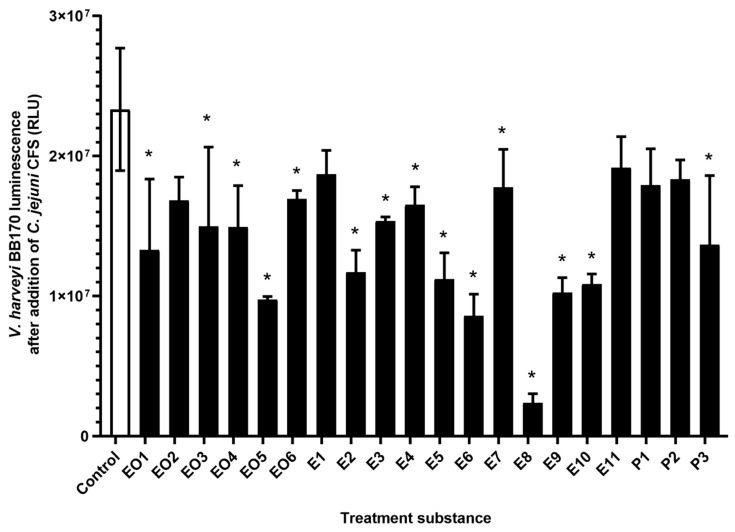
Luminescence of *V. harveyi* BB170 cells (relative fluorescent units; RLU) after addition of *C. jejuni* 11168 cell-free supernatants from untreated *C. jejuni* cultures (control, empty bar) and *C. jejuni* cultures treated with essential oils (EO1–6), ethanolic extracts (E1–11), and pure compounds (P1–3) (full bars). Data are means ± standard deviation. *, *p* < 0.05 versus control (Student’s *t*-tests).

**Figure 3 microorganisms-08-00104-f003:**
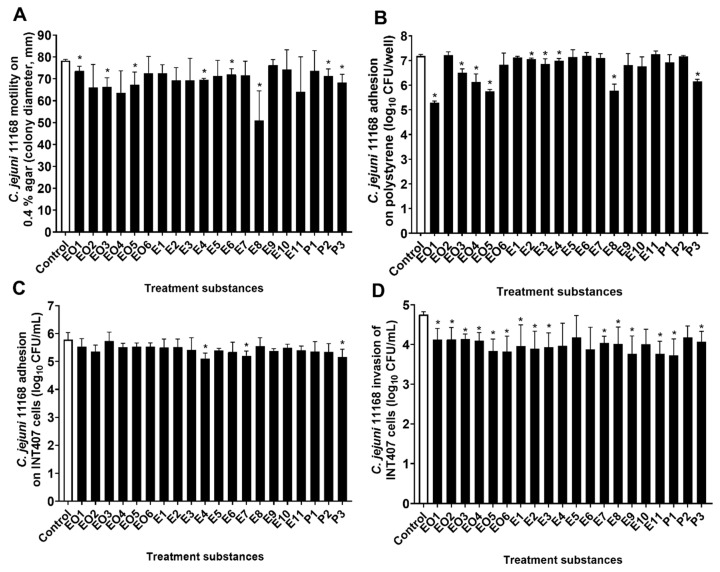
*C. jejuni* 11168 motility on 0.4% agar (**A**); colony diameter (mm), adhesion to polystyrene surfaces (**B**); log10 CFU/well, adhesion to (**C**) and invasion of (**D**) INT407 cells (CFU/mL) without (control, empty bars) and with treatments with essential oils (EO1–6), ethanolic extracts (E1–11), and pure compounds (P1–3) (full bars). Data are means ± standard deviations. * *p* < 0.05 versus control.

**Table 1 microorganisms-08-00104-t001:** Inhibition of *C. jejuni* 11168 QS, motility, adhesion to polystyrene surfaces, and adhesion to and invasion of INT407 cells, as calculated from the cultures treated with essential oils (EO1–6), ethanolic extracts (E1–11), and pure compounds (P1–3) compared to the control, presented as %inhibition ± standard deviation.

Treatment	Inhibition (%) for Treated Cultures Compared to Control Culture			
Code	QS	Motility	Adhesion to Polystyrene Surfaces	Invasion of INT407 Cells
EO1	51 ± 11.6	6 ± 0.1	99 ± 0.9	76 ± 4.3
EO2	37 ± 1.7	16 ± 2.0	−10 ± 0.1	76 ± 4.5
EO3	44 ± 9.4	15 ± 0.8	79 ± 1.6	76 ± 2.0
EO4	44 ± 4.9	19 ± 2.4	91 ± 3.8	78 ± 3.1
EO5	66 ± 0.7	14 ± 1.0	96 ± 1.2	88 ± 5.6
EO6	36 ± 0.7	7 ± 0.6	56 ± 3.2	88 ± 7.3
E1	29 ± 1.3	7 ± 0.3	13 ± 0.1	84 ± 9.8
E2	58 ± 4.5	11 ± 0.8	26 ± 0.1	86 ± 8.5
E3	43 ± 0.4	11 ± 1.4	54 ± 1.4	85 ± 7.0
E4	38 ± 1.5	11 ± 0.2	36 ± 0.4	84 ± 9.8
E5	60 ± 5.2	9 ± 0.7	11 ± 0.4	73 ± 7.9
E6	71 ± 6.9	8 ± 0.2	−1 ± 0.1	87 ± 10.2
E7	33 ± 2.8	9 ± 0.6	15 ± 0.3	81 ± 2.8
E8	96 ± 15.7	35 ± 7.6	96 ± 3.7	82 ± 7.1
E9	64 ± 3.5	3 ± 0.1	57 ± 3.2	90 ± 8.8
E10	61 ± 2.3	5 ± 0.5	62 ± 2.9	82 ± 7.1
E11	27 ± 1.5	18 ± 3.8	−18 ± 0.3	90 ± 6.3
P1	32 ± 2.3	6 ± 0.6	44 ± 1.6	91 ± 8.2
P2	31 ± 1.2	9 ± 0.3	2 ± 0.1	73 ± 4.0
P3	50 ± 10.4	13 ± 0.6	91 ± 0.9	79 ± 4.2

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
