# Peer review of "Modulation of Campylobacter jejuni Motility, Adhesion to Polystyrene Surfaces, and Invasion of INT407 Cells by Quorum-Sensing Inhibition"

_microorganisms, 2020, doi:10.3390/microorganisms8010104_

Round 1
Reviewer 1 Report
The paper was well written and the results were well presented, although the discussions could be further improved.
Specific comments:
As the complementation of the C. jejuni luxS mutant wasn't carried out, avoid using the "We thus confirm that" in the last sentence in the Abstract. 3.4: Results for C.jejuni 11168ΔluxS is missing Line 350-352: delete "minor but", "significant reduction of" and the last "significant reduction in". Line 361-365: delete Line 366-372: Shorten it, such as " The responses of C. jejuni 11168 and C.jejuni 11168ΔluxS to 20 natural formulations were compared, using the concentrations of 0.25xMIC to ensure they did not influence C. jejuni growth". Line 442-445: rewritten to make it shorter
Author Response
Response to Reviewer 1 Comments
Point 1: The paper was well written and the results were well presented, although the discussions could be further improved.
Response 1: Thank you for your observations. Modifications have been made in the discussion (L360-363; L372-387; L458-452).
Point 2: As the complementation of the C. jejuni luxS mutant wasn't carried out, avoid using the "We thus confirm that" in the last sentence in the Abstract.
Response 2: Thank you for this observation. Changes have been made in L28.
Point 3: 3.4: Results for C.jejuni 11168ΔluxS is missing
Response 3: To correct this mistake, a sentence in L305-306 has been added.
Point 4: Line 350-352: delete "minor but", "significant reduction of" and the last "significant reduction in".
Response 4: The proposed deletion has been carried out (L360-363).
Point 5: Line 361-365: delete
Response 5: The sentence has been removed.
Point 6: Line 366-372: Shorten it, such as " The responses of C. jejuni 11168 and C.jejuni 11168ΔluxS to 20 natural formulations were compared, using the concentrations of 0.25xMIC to ensure they did not influence C. jejuni growth".
Response 6: The sentence has been shortened in L384-387.
Point 7: Line 442-445: rewritten to make it shorter
Response 7: The sentence has been rewritten in L460-464.
Reviewer 2 Report
General comments
The current article is a continuation of several studies carried out by the authors about the main bacterial foodborne pathogen such as Campylobacter jejuni. Campylobacteriosis is the most frequently reported zoonotic disease in humans in Europe over last decade and the search of new targets for C. jejuni inhibition and new alternatives to use of antibiotics for its control are necessary. In my opinion, the study is well designed and the methodology is sound, and the results are interesting and novel, in part. The results and discussion is well formulated and the limitations of the study are carefully analysed.
Review report
Title: Concise and reflect the content of the article.
Suggestions:
Polystyrene surfaces or abiotic surfaces, consider for the rest of the manuscript to not confuse with adhesion to cells. The term “inhibition” is excessively absolute. Is the QS totally inhibited?
Abstract: It is brief and describes the purpose of the work and the major results.
Suggestions:
“…treatment of C. jejuni with 6 essential oils, 11 ethanolic extracts, and 3 pure compounds”.
Introduction: This section is clear and well organised. Reflect the importance of C. jejuni as a foodborne pathogen and the demand to obtain new alternatives to its control. The authors explain the bacterial quorum sensing (QS) and its signalling regulation. Finally, the authors propose to study th influence of QS inhibition on C. jejuni motility, adhesion to polystyrene, and adhesion to and invasion of INT407 epithelial cells.
Material and methods: Properly describes the methodology used. Only minor modifications should be included:
Please, include the origin of the C. jejuni 11168 and its mutant strain. Did C. jejuni grow in only 24h? Amazing… L118-121. The authors used pre- and post-distilleted extracts but not explain why did it. L142-143. OD600 0.1 what is the equivalent in CFU/mL? Please, include this data. After 16 h of culture incubation, which is the inoculum in CFU/mL? The used 96-well polystyrene microplates are treated? L203 and 213. fetal bovine serum. Why the authors lysed the cells for determination of C. jejuni adherence? If the cells are lysed the final bacterial counting include adhered and invaded cells.
Results: The presentation of the results is clear and reflect the main results. Comments and corrections:
Have authors a hypothesis why the invasion in mutant strain is affected but no the adherence? Both virulence factors are strongly related. L256 and 273. Correct, means ± standard deviation. Table 1. Data column about jejuni 11168 adhesion to INT407 cells is missing. Why is not in the data included the standard deviation? Please, include the SD. Negative inhibition value (%) for adhesion to polystyrene? It would be a zero inhibition, isn’t it? Figure 3. It is difficult to obtain the absolute data from this figure, I recommend to the authors to create a table that includes the mean and standard deviation of figures 3A, 3B, 3C, and 3D. When authors explain in L304-305, L315-316 or L319-320 the log reduction is not easy to view in Figure 3. Please, transform the figure into a table. L300-301. Please, correct the sentence. The adhesion of the wild‐type was “significantly” reduced by nine of these treatments (n = 9/20)… The invasion of the INT407 cells by C. jejuni 11168 was significantly modified by 11 natural extracts (n = 11/20, Fig. 3D), is not correct the proportion n = 15/20 included by the authors. Asterisk only appear in 11 treatments. Please, check this information. L321-322. EO6, E2, and E3 treatments had significant effect according to Fig. 3D. Significance asterisk appear in the full bars of these treatments. Please, check this information.
Discussion:
Correct the sentence: …polystyrene, and “adhesion to and invasion” of human INT407 epithelial cells. Correct the order of the references cited, Holmes et al [17] and then Quiñones [18]. L354 and L358. Reference [18], the correct surname is Quiñones, please include the correct name. L375-376. This sentence must include the reference to the figure 2. The motility of C. jejuni 11168 was significantly reduced by only “eight” of these 20 treatments… Please, check this information, according to Fig. 3A eight extracts were significantly actives. (n = 11/20) not (n = 15/20). Authors must include some information and references about the main composition of the most active extracts E8 (roseroot). L433-435. Extract E11 seems to have weaker effect then P2 compound, isn’t it? The authors not explain if is most effective the lack of LuxS or the treatment whit the natural extract?
References: The reference section is appropriate and completely updated.
Some references about the composition of the most active extract (E8) must be include in the revised manuscript.
Author Response
Response to Reviewer 2 Comments
Title: Concise and reflect the content of the article.
Point 1: Suggestions: Polystyrene surfaces or abiotic surfaces, consider for the rest of the manuscript to not confuse with adhesion to cells. The term “inhibition” is excessively absolute. Is the QS totally inhibited?
Response 1: The term “polystyrene” has been exchanged for “polystyrene surfaces” throughout the manuscript. The term “inhibition” has been changed where appropriate. QS is not totally inhibited but the influence of the treatments is shown as % of inhibition.
Abstract: It is brief and describes the purpose of the work and the major results.
Point 2: Suggestions: “…treatment of C. jejuni with 6 essential oils, 11 ethanolic extracts, and 3 pure compounds”.
Response 2: The suggestion has been taken into account and changes were made in L20-21.
Introduction: This section is clear and well organised. Reflect the importance of C. jejuni as a foodborne pathogen and the demand to obtain new alternatives to its control. The authors explain the bacterial quorum sensing (QS) and its signalling regulation. Finally, the authors propose to study th influence of QS inhibition on C. jejuni motility, adhesion to polystyrene, and adhesion to and invasion of INT407 epithelial cells.
Material and methods: Properly describes the methodology used. Only minor modifications should be included:
Point 3: Please, include the origin of the C. jejuni 11168 and its mutant strain.
Response 3: Additional information has been added in L99-100.
Point 4: Did C. jejuni grow in only 24h? Amazing…
Response 4: Yes. We find that most of our C. jejuni lab strains grow to a sufficient amount after 24 h when revitalizing them.
Point 5: L118-121. The authors used pre- and post-distilleted extracts but not explain why did it.
Response 5: As the chemical composition of pre- and post-distilled extracts if often very different we considered them as separate extracts of the same plant. For further work this information of differences between pre- and post-distilled extracts will be important, but in this study they serve as separate extracts.
Point 6: L142-143. OD600 0.1 what is the equivalent in CFU/mL? Please, include this data.
Response 6: The information has been added in L146.
Point 7: After 16 h of culture incubation, which is the inoculum in CFU/mL?
Response 7: The information has been added in L154.
Point 8: The used 96-well polystyrene microplates are treated?
Response 8: The plates were non-treated. The information has been added in L190.
Point 9: L203 and 213. fetal bovine serum.
Response 9: Thank you for the observation. The error has been corrected in L206 and L216.
Point 10: Why the authors lysed the cells for determination of C. jejuni adherence? If the cells are lysed the final bacterial counting include adhered and invaded cells.
Response 10: Yes, by lysing the cells the number of both is determined. Further, with the addition of gentamicin only the number of invaded cells is determined and then this number is subtracted from the overall number to get adhered cells. Additional explanation was added in L229-231.
Results: The presentation of the results is clear and reflect the main results. Comments and corrections:
Point 11: Have authors a hypothesis why the invasion in mutant strain is affected but no the adherence? Both virulence factors are strongly related.
Response 11: At this point, without further experiments, we do not feel that giving an explanation for this phenomenon would be wise. Although, it is an interesting result and will be investigated in the future.
Point 12: L256 and 273. Correct, means ± standard deviation.
Response 12: Corrected (L262 and L279).
Point 13: Table 1. Data column about jejuni 11168 adhesion to INT407 cells is missing.
Response 13: The data for adhesion reduction was not included as statistical significance was not achieved and no correlation was found between QS and adhesion on INT407, thus we did not feel that the data representation as such was applicable here.
Point 14: Why is not in the data included the standard deviation? Please, include the SD.
Response 14: Standard deviation were added in Table 1.
Point 15: Negative inhibition value (%) for adhesion to polystyrene? It would be a zero inhibition, isn’t it?
Response 15: This is correct although a negative value indicated somewhat increased adhesion in treated culture compared to control and we feel that this information is important too and we would leave it as such.
Point 16: Figure 3. It is difficult to obtain the absolute data from this figure, I recommend to the authors to create a table that includes the mean and standard deviation of figures 3A, 3B, 3C, and 3D. When authors explain in L304-305, L315-316 or L319-320 the log reduction is not easy to view in Figure 3. Please, transform the figure into a table.
Response 16. Thank you for the feedback. We do feel that the figure is a valuable part of the manuscript and would keep it. Instead, we added this information as a Supplementary Table S2 in the supplementary material.
Point 17: L300-301. Please, correct the sentence. The adhesion of the wild‐type was “significantly” reduced by nine of these treatments (n = 9/20)…
Response 17: It was corrected (L308).
Point 18: The invasion of the INT407 cells by C. jejuni 11168 was significantly modified by 11 natural extracts (n = 11/20, Fig. 3D), is not correct the proportion n = 15/20 included by the authors. Asterisk only appear in 11 treatments. Please, check this information.
Response 18: Thank you for this observation. An error occurred in the data representation and it was corrected in the figure and in text (L330-333).
Point 19: L321-322. EO6, E2, and E3 treatments had significant effect according to Fig. 3D. Significance asterisk appear in the full bars of these treatments. Please, check this information.
Response 19: The error was corrected in the figure and in text (L330-333).
Discussion:
Point 20: Correct the sentence: …polystyrene, and “adhesion to and invasion” of human INT407 epithelial cells.
Response 20: It has been corrected in L360.
Point 21: Correct the order of the references cited, Holmes et al [17] and then Quiñones [18].
Response 21: Corrected.
Point 22: L354 and L358. Reference [18], the correct surname is Quiñones, please include the correct name.
Response 22: It has been corrected in L365 and L370.
Point 23: L375-376. This sentence must include the reference to the figure 2.
Response 23: The reference was added in L391.
Point 24: The motility of C. jejuni 11168 was significantly reduced by only “eight” of these 20 treatments… Please, check this information, according to Fig. 3A eight extracts were significantly actives.
Response 24: It has been corrected in L406.
Point 25: (n = 11/20) not (n = 15/20).
Response 25: N=15/20 is correct.
Point 26: Authors must include some information and references about the main composition of the most active extracts E8 (roseroot).
Response 26: More information has been added in L448-451.
Point 27: L433-435. Extract E11 seems to have weaker effect then P2 compound, isn’t it?
Response 27: Overall P2 has weaker effect. The sentence has been corrected (L452).
Point 28: The authors not explain if is most effective the lack of LuxS or the treatment whit the natural extract?
Response 28: For each phenotype we have compared the effect of the luxS knockout mutation. We can not draw only one conclusion as there are differences in effect in each system.
References: The reference section is appropriate and completely updated.
Point 29: Some references about the composition of the most active extract (E8) must be include in the revised manuscript.
Response 29: The reference describing the chemistry of the roseroot extract is included as Alperth et al., 2019. Additional text has been added in the discussion L448-451.